# Open Reduction in Subtrochanteric Femur Fractures Is Not Accompanied by a Higher Rate of Complications

**DOI:** 10.3390/medicina57070659

**Published:** 2021-06-27

**Authors:** Tom Knauf, Daphne Eschbach, Benjamin Buecking, Matthias Knobe, Juliane Barthel, Katherine Rascher, Steffen Ruchholtz, Rene Aigner, Carsten Schoeneberg

**Affiliations:** 1Center for Orthopaedics and Trauma Surgery, University Hospital Giessen and Marburg, 35039 Marburg, Germany; eschbach@med.uni-marburg.de (D.E.); barthelj@med.uni-marburg.de (J.B.); ruchholt@med.uni-marburg.de (S.R.); aignerr@med.uni-marburg.de (R.A.); 2Center for Orthopaedics and Trauma Surgery, DRK-Kliniken Nordhessen, 34121 Kassel, Germany; buecking@drk-nh.de; 3Department of Orthopaedic and Trauma Surgery, Lucerne Cantonal Hospital, 6004 Lucerne, Switzerland; Matthias.knobe@luks.ch; 4AUC, Akademie der Unfallchirurgie GmbH, 80639 Munich, Germany; katherine.rascher@auc-online.de; 5Department of Orthopedic and Emergency Surgery, Alfried Krupp Hospital, 45131 Essen, Germany; carsten.schoeneberg@krupp-krankenhaus.de

**Keywords:** subtrochanteric fracture, reduction, geriatric patient, hip fracture

## Abstract

*Background**and Objectives*: Hip fractures are among the most typical geriatric fractures. Subtrochanteric fractures are considered difficult to treat, and, to date, there is no consensus on the optimal surgical treatment. *Materialis**and Methods*: We analyzed data from the Registry for Geriatric Trauma, which includes patients ≥ 70 years old with hip fractures or periprosthetic fractures requiring surgery (21,734 patients in 2017–2019). For this study, we analyzed only the subgroup of patients with a subtrochanteric fracture. We analyzed the difference between closed and open surgical methods on a range of outcomes, including mortality, mobility, length of acute hospital stay, and the need for surgical revisions. *Results*: A total of 506 patients with subtrochanteric fractures were analyzed in this study. The median age was 85 years (interquartile range of 81–89). About 21.1% (*n* = 107) were operated on with a closed technique, 73.3% (*n* = 371) with open reduction without using a cerclage, and 5.53% (*n* = 28) with open reduction with the additional use of one or more cerclage wires. A total of 3.56% (*n* = 18) of the patients had complications requiring operative revision, most commonly soft tissue interventions (open vs. closed reduction—3.26% vs. 4.67%) (*p* = 0.687). Patients treated with open reduction were significantly more mobile 7 days after surgery (*p* = 0.008), while no significant effects on mortality (*p* = 0.312), length of hospital stay (*p* = 0.968), or surgical complications (*p* = 0.687) were found. *Conclusion*: Proper reduction is the gold standard practice for successful union in subtrochanteric fractures. This study shows that open reduction is not associated with a higher complication rate but does lead to increased mobility 7 days after operation. Therefore, in case of doubt, a good reduction should be aimed for, even using open techniques.

## 1. Introduction

Due to demographic changes, the treatment of geriatric patients is gaining more attention nowadays. Hip fractures remain the most typical fragility fractures in geriatric patients. Despite great efforts in optimizing perioperative treatment and long-term rehabilitation programs, per- and subtrochanteric fractures are associated with high morbidity and mortality, reduced quality of life, and a loss of postoperative autonomy [1,2,3,4]. Of all proximal femur fracture types, subtrochanteric fractures account for 8–34% of the trochanteric fractures [3,5] and are associated with the worst outcomes [3,6].

Intramedullary stabilization is the favored surgical procedure for these fractures [3,7]. However, it is not clear which type of reduction (open vs. closed) is best [8,9,10,11,12]. Some studies have found that open reduction leads to a more anatomic reposition and lower malunion rates [2,8,9,11]. In contrast, other authors report higher non-union rates and complications regarding soft tissue healing with open reduction [13,14]. A retrospective analysis by Shukla et al. showed no increase of complications with regard to wound infections, recovery rates, and some other minor complications, independent of the greater intraoperative soft tissue trauma following open reduction [8]. Codesido et al. support these findings. Despite longer operation times for patients that were treated with open reduction, they found reduced lengths of hospital stay by 2 days and showed an improved health-related quality of life (EQ-5D) compared to closed reduction patients [15]. However, the results of the current literature are inconsistent [13,16]. Therefore, we analyzed the data of the Registry for Geriatric Trauma (ATR-DGU) to study the differences of closed vs. open reduction of subtrochanteric femoral fractures in an orthogeriatric treated collective. The primary outcome parameter was the occurrence of complications with a need for surgical revision. Secondary outcome parameters were mortality, mobility, and length of hospital stay. We hypothesized that the treatment of subtrochanteric fractures with open reduction does not lead to higher rates of surgical revisions.

## 2. Materials and Methods

### 2.1. Data Source

The ATR-DGU was founded by the German Trauma Society (DGU) in 2016. The ATR-DGU encompasses standardized, pseudonymized documentation of geriatric patients with a proximal femur fracture requiring surgery. The infrastructure for documentation, data management, and data analysis is provided by the Academy for Trauma Surgery (AUC—Akademie der Unfallchirurgie GmbH). The scientific oversight is carried out by the Working Committee on Geriatric Trauma Registry (AK ATR) of the DGU. Participating centers transmit pseudonymized patient data via a web-based application into a central database. Approval for scientific data analysis from the ATR-DGU is granted via a peer-review process in accordance with the publication guidelines laid down by the AK ATR [17]. Currently, hospitals from Germany, Switzerland, and Austria contribute to the ATR-DGU, with a total of nearly 25,000 cases from about 100 hospitals. The present study is in accordance with the publication guidelines of the ATR-DGU and registered as ATR-DGU project ID 2020-007.

### 2.2. Patient Inclusion

The inclusion criteria of the ATR-DGU are hip fractures with a need for surgery and the age of 70 years or older. Patients who died before surgery were excluded. The data was collected during 5 relevant time periods: admission, preoperatively, during the first postoperative week, at discharge, and during an optional follow-up on the 120th postoperative day.

### 2.3. Statistical Analyses

Continuous variables were expressed as the median with an interquartile range (IQR), and categorical variables as counts and percentages. The variables were compared between the groups (open vs. closed reduction) using Wilcoxon–Mann–Whitney U-tests for continuous variables and chi-squared tests for categorical variables. A binary logistic regression was performed to further examine the influence of the open vs closed reduction on walking ability on the 7th post-operative day. As is common with registry data, not all information was available for each patient. All analyses were performed in R version 4.0.2.

## 3. Results

Between 2017 and 2019, 21,734 patients were entered into the ATR-DGU register. A total of 841 of those patients suffered from subtrochanteric fracture. Of these patients, 786 were treated with an intramedullary nail. For 280 patients, there was no data on the operative treatment (open vs. closed reduction). After the exclusion of all incomplete datasets, it was possible to analyze the data from 506 patients from 75 hospitals. We further divided the patients into two groups: closed reduction (*n* = 107) and open reduction (*n* = 399).

The 120-day follow-up data was available for roughly one-third of these patients (*n* = 194).

### 3.1. Baseline Characteristics

The median age was 85 years (interquartile range (IQR) of 81–89) and 74.1% (*n* = 375) of patients were female. About 76.4% (*n* = 367 patients) of the patients had an American-Scoiety of Anaesthesiologists (ASA)-score ≥ 3, indicating severe systemic disease, and almost half of the patients took anticoagulants prior to their fracture (49.7%; *n* = 242). The median length of stay in the hospital was 17 days (IQR of 11–22). A total of 21.1% (*n* = 107) of patients were treated with closed reduction, 73.3% (*n* = 371) with open reduction without wire cerclage, and 5.53% (*n* = 28) with open reduction and the use of a wire cerclage. The baseline characteristics and patient counts for each parameter can be found in Table 1.

### 3.2. Primary Outcome Parameter

“Complications requiring surgical revision” was defined as the primary outcome parameter. A total of 3.56% (*n* = 18) of the patients suffered from such complications during the initial hospital stay. Soft tissue interventions were the most common surgical revisions performed (Table 1). Comparing open vs. closed reduction, 3.26% of the patients treated using open reduction required a revision surgery versus 4.67% of the patients treated with closed reduction, but this difference was not statistically significant (*p* = 0.687) (Table 2; Figure 1). No significant difference regarding complications leading to revision surgery was seen in the follow-up data (*p* = 1) (Table 3).

### 3.3. Secondary Outcome Parameters

Whether a fracture was treated with open or closed reduction had a significant influence on the change in walking ability (pre-fracture to seven days post-surgery) (*p* = 0.008; Table 2). A total of 38.3% of the patients treated with open reduction had at least the same level of mobility compared to the pre-fracture level, while only 22.9% of the patients treated with closed reduction had that level of mobility. Patients treated with open reduction had a mortality rate at the 120-day follow-up of 8.4%. vs. 0% in the closed reduction group; however, due to the small sample size, this difference was not statistically significant (*p* = 0.075).

The positive influence of open reduction on walking ability on the seventh day after surgery was confirmed in a multivariate logistic regression adjusted for age, gender, ASA score, and mobility before the fracture (odds ratio—2.07; 95% confidence interval (1.25; 3.42); *p* = 0.004).

## 4. Discussion

In order to ensure optimal fracture care, visualization of the fracture is important. While often a 2-plane x-ray is sufficient, there are cases where computer tomography (CT) with 3D reconstruction may be used for better visualization of the fracture [18,19]. It is broadly recognized that proper reduction is necessary for the successful union in subtrochanteric fractures [2,9,11]. Higher non-union rates and complications regarding soft tissue healing are feared complications of open reductions [13,14]. The aim of this study was to evaluate whether the reduction type (open vs. closed) of subtrochanteric femoral fractures influenced surgical revision rates, mortality, mobility, or length of hospital stay for geriatric hip fracture patients.

### 4.1. Primary Outcome Parameter

In order to provide a suitable reduction of subtrochanteric fractures, different techniques have been described, from closed reduction to mini-open procedures and open reduction using cerclage wires and cables. Some authors hesitate to perform open reduction because of prolonged wound healing, soft tissue complications, and the risk of nonunion [13,14]. They recommend open reduction only after all other closed reduction options have been exhausted [11]. In our study, the type of reduction (open vs. closed) showed no significant difference with regard to complications (open reduction: 3.3% vs. closed reduction: 4.7%). When performed properly, open reduction might be one key point to avoid malalignment without the risk of further soft tissue complications. Krappinger et al. studied risk factors for nonunion after intramedullary nailing of subtrochanteric fractures. In their study, neither open reduction nor the use of wire cerclage were significant risk factors for nonunion [10]. This is in contrast to malalignment, which has been shown to be a risk factor for nonunion [8,9,10]. Kasha et al. concluded that regardless of the method chosen, the key point to reduce the risk of complications is adequate reduction [11]. Trikha et al. concluded, in their retrospective analysis, minimally invasive cerclage wire application to be beneficial for anatomical reconstruction in difficult subtrochanteric fractures [12]. Therefore, the high rate of open reduction is not surprising. A total of 78.9% of the fractures of patients included in the ATR-DGU were treated with open reduction. This rate is quite high compared with rates described by other authors (48.1%) [20]. Contrary to this, only 5.5% of the fractures were treated with an additional wire cerclage. This rate is lower compared to the earlier literature (14.8%) [20]. Differences may be due to the different study collectives that were described. Our study only focused on geriatric trauma patients, whereas age was not an exclusion criterion in the other studies [20]. Additionally, all of our patients were treated in an orthogeriatric setting [21].

### 4.2. Secondary Outcome Parameters

According to Dubljanin et al., the functional level at discharge is the main determinant of 1-year mortality for hip fracture patients [22]. Therefore, this may serve as a screening tool to predict the further outcome for these patients. Looking at the mobility of hip fracture patients in our study, on the seventh day after surgery, 69.2% of the patients had some walking ability. A closer look at the change in mobility of these patients on the seventh day after surgery showed that 38.3% of the patients treated with open reduction had at least the same level of mobility compared to the pre-fracture level, while only 22.9% of the patients treated with closed reduction had that level of mobility. This increased mobility of patients treated using open reduction is quite remarkable when taking into account that only in 66% of the patients was full weight-bearing allowed (75.7% of the patients with closed reduction). None of the other parameters analyzed (mortality, length of hospital stay, surgical complication, and discharge location) showed a significant difference between the open and closed reduction groups during the initial stay or the follow-up. The results of the follow-ups must be viewed with caution, as they were evaluated in only one-third of the patients. Additionally, it must also be critically mentioned here that the follow-up was relatively short at 120 days. It is possible that an association between improved mobility and decreased mortality would be seen if the follow-up was conducted later. In contrast, we found a tendency for higher mortality in the open reduction group (8.4%. vs. 0% in the closed reduction group). Therefore, we can draw no conclusions about the influence of reduction type on mortality.

### 4.3. Strengths and Limitations of the Study

As a limitation of this study, it has to be mentioned that not all items were collected from all patients, which is unfortunately common in registry studies. Additionally, the register lacks information regarding the quality of reduction and the quality of the union after 120 days, which would have been interesting in evaluating the postoperative result. As already mentioned above, the low follow-up rate is a further limiting factor.

Whether or not an additional cerclage had an impact on the outcome parameters would also be interesting to evaluate. Unfortunately, only 5.5% of the patients were treated with an additional cerclage, making this analysis not possible. Therefore, patients treated with an additional cerclage were included in the “open-reduction” group.

Unfortunately, we were not able to reproduce which criteria led the surgeons to perform an open reduction, since this parameter was not collected in the registry study.

Nevertheless, the strength of this study is the large number of patients suffering from this special type of fracture. Patients from multiple centers all over Germany, Austria, and Switzerland were included in this study, which has to be mentioned as an additional strength of this study. Furthermore, the requirement that data was only obtained from orthogeriatric centers makes this collection a very uniform one and reduces the multiple confounding factors that may make the evaluation of such treatment data difficult.

## 5. Conclusions

Subtrochanteric fractures remain a rare subtype of proximal femoral fractures in the geriatric trauma population. Proper reduction is necessary for the successful healing of subtrochanteric fractures. Some surgeons are hesitant to perform open reduction due to concerns about soft tissue healing and other complications. Our study showed that open reduction was not accompanied by a higher complication rate, mortality rate, nor an increased length of hospital stay. Seven days after surgery, patients treated with open reduction had better mobility, which is an important step in the recovery of geriatric patients. There might be situations in which a less than optimal reduction can be accepted in favor of a more minimally invasive operation. However, in the case of a subtrochanteric fracture, a critical evaluation should be made. In case of doubt, a good reduction should be aimed for, even if that means performing an open reduction.

## Figures and Tables

**Figure 1 medicina-57-00659-f001:**
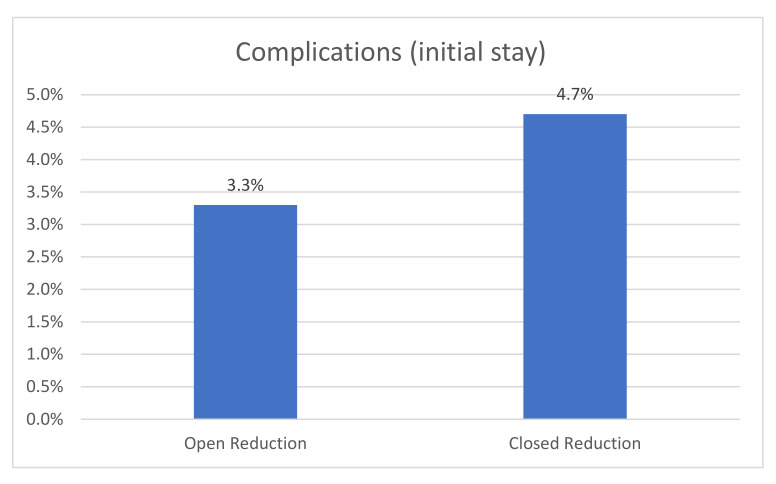
The rate of complications after open vs. closed reduction.

**Table 1 medicina-57-00659-t001:** Baseline characteristics.

Patient Characteristics	All (*n* = 506)
**Baseline Data—Initial Stay**	
**Age (n)**	*n* = 506
**Median (Interquartile Range) (IQR)**	85 (81, 89)
**Gender**	*n*= 506
Female	74.1% (*n* = 375)
**ASA Score**	***n*** **= 480**
1	0.417% (*n* = 2)
2	23.1% (*n* = 111)
3	67.9% (*n* = 326)
4	8.12% (*n* = 39)
5	0.417% (*n* = 2)
**Type of Surgery (Subtrochanteric Fracture)**	*n* = 506
Intramedullary nail open reduction without wire cerclage	73.3% (*n* = 371)
Intramedullary nail open reduction with wire cerclage	5.5% (*n* = 28)
Intramedullary nail closed reduction	21.1% (*n* = 107)
**Length of Stay (Median/IQR) (*n* = 463)** **(Survivors)**	17.04 (11.06, 22.08)
**Anticoagulation on Admission (*n* = 487)**	
No anticoagulation	50.3% (*n* = 245)
Vitamin K antagonist	18.2% (*n* = 44)
Acetylsalicylic acid	49.6% (*n* = 120)
Other thrombocyte aggregation inhibitors	7.02% (*n* = 17)
Direct thrombin inhibitor (Dabigatran)	3.72% (*n* = 9)
Direct factor Xa inhibitor (Rivaroxaban, Apixaban)	22.3% (*n* = 54)
Heparin	2.07% (*n* = 5)
Other	2.07% (*n* = 5)
**Surgical Complication (during initial stay)**	3.56% (*n* = 18)
Removal of implant or osteosynthesis	11.1% (*n* = 2)
Revision of osteosynthesis	22.2% (*n* = 4)
Soft tissue intervention	66.7% (*n* = 12)
Others	16.7% (*n* = 3)
**Mortality**	(*n* = 505)
During initial stay	7.52% (*n* = 38)
**Length of Stay (Median/IQR) (*n*=)** **(Survivors)**	*n* = 463 17.04 (11.06, 22.08)
**Discharge After Hospital (initial stay)**	*n* = 461
Home	22.28% (*n* = 105)
Nursing home	29.9% (*n* = 138)
Geriatric rehabilitation	41.6% (*n* = 192)
Clinic for follow-up treatment	3.90% (*n* = 18)
Other (different hospital, different department, other)	1.74% (*n* = 8)
***Baseline data—follow-up***	
**Surgical Complication**	
Yes	1.38% (*n* = 7)
Removal of implant or osteosynthesis	42.9% (*n* = 3)
Revision of osteosynthesis	14.3% (*n* = 1)
Conversion in total hip arthroplasty	14.3% (*n* = 1)
Girdlestone	14.3% (*n* = 1)
Soft tissue intervention	14.3% (*n* = 1)
Others	28.6% (*n* = 2)
**Mortality**	
Yes	2.57% (*n* = 13)
**Current Location**	*n* = 140
Home	72.9% (*n* = 102)
Nursing home	25.7% (*n* = 36)
Hospital	0.714% (*n* = 1)
Other	0.714% (*n* = 1)

ASA-Score:American-Society of Anaesthesiologists-Score.

**Table 2 medicina-57-00659-t002:** Open vs. closed reduction (initial stay).

	Open Reduction (*n* = 399)	Closed Reduction (*n* = 107)	*p*
***Baseline Data***			
**Age *n***	399	107	0.327 ^*^
Median (Interquartile Range) (IQR)	85 (81; 89)	84 (80; 89)	
**Gender**			0.233 ^+^
Female	301(75.4%)	74 (69.2%)	
Male	98 (24.6%)	33 (30.8%)	
**ASA Score**			0.198 ^+^
1	1 (0.3%)	1 (1.0%)	
2	92 (24.4%)	19 (18.4%)	
3	256 (67.9%)	70 (68.0%)	
4	26 (6.9%)	13 (12.6%)	
5	2 (0.5%)		
**Anticoagulation on Admission**			0.088 ^+^
No Anticoagulation	48.2% (*n* = 185)	58.3 ((*n* = 60)	
**Walking Ability Before Fracture**			
Without aids/forearm crutches	168 (46.2%)	52 (52%)	0.538 ^+^
Walker	128 (35.2%)	28 (28%)	
Only at home	58 (15.9%)	16 (16%)	
No walking possible	10 (2.8%)	4 (4%)	
**Full Weight Bearing of the Fracture is Allowed**			0.073
Yes	66.0% (*n* = 262)	75.7% (*n* = 81)	
***Primary Outcome Parameter***			
**Surgical Complication**			0.687
**Yes**	3.26% (*n* = 13)	4.67% (*n* = 5)	
Soft tissue intervention	2.5% (*n* = 10)	1.8% (*n* = 2)	
Removal of Implant or osteosynthesis	0.3% (*n* = 1)	0.9% (*n* = 1)	
Revision of osteosynthesis	0.5% (*n* = 2)	1.8% (*n* = 2)	
Periosteosynthetic fracture	0% (*n* = 0)	0.9% (*n* = 1)	
Others	0.8% (*n* = 3)	0% (*n* = 0)	
***Secondary Outcome Parameter***			
**Discharge After Hospital (initial stay) (Survivors)**	*n* = 366	*n* = 95	*p* = 0.846
Home	23.5% (*n* = 86)	20% (*n* = 19)	
Nursing home	30.3% (*n* = 111)	28.4% (*n* = 27)	
Geriatric rehabilitation	40.4% (*n* = 148)	46.3% (*n* = 44)	
Clinic for follow-up treatment	4.10% (*n* = 15)	3.16% (*n* = 3)	
Other	1.64% (*n* = 6)	2.11% (*n* = 2)	
**Mortality**	6.78% (*n* = 27)	10.3% (*n* = 11)	0.312
**Change in Walking Ability (pre-fracture to 7 days post-op)**			0.008 ^+^
Worse	61.8% (*n* = 218)	77.1% (*n* = 74)	
No change	28.9% (*n* = 102)	20.8% (*n* = 20)	
Better	9.4% (*n* = 33)	2.1% (*n* = 2)	
**Lengths of stay (days) (Survivors)**	*n* = 367	*n* = 96	0.968 ^^^
Median (IQR)	17 (11, 22.1)	16.6 (12, 21.6)	

^*^ Mann–Whitney U test; ^+^ Chi-square test; ^^^ Wilcox test.

**Table 3 medicina-57-00659-t003:** Open reduction vs. closed reduction (follow-up).

	Open Reduction	Closed Reduction	
Follow-Up Conducted	38.8% (*n* = 155)	36.4% (*n* = 39)	
***Primary Outcome Parameter***			
**Surgical Complication**	*n* = 130	*n* = 36	*p* = 1
**Yes**	3.9% (*n* = 5)	5.6% (*n* = 2)	
Soft tissue intervention	*n* = 0	*n* = 1	
Removal of implant or osteosynthesis	*n* = 2	*n* = 1	
Revision of osteosynthesis	*n* = 1	*n* = 0	
Conversion in HTEP	*n* = 1	*n* = 0	
Girdlestone	*n* = 1	*n* = 0	
Others	*n* = 1	*n* = 1	
***Secondary Outcome Parameter***			
**Mortality**	8.39% (*n* = 13 additional deaths in FU Period)	0% (*n* = 0)	0.075
**Change in Walking Ability (pre-fracture to 120-day follow-up)**			0.82 ^+^
Worse	65 (60.2%)	19 (59.4%)	
No change	39 (36.1%)	11 (34.4%)	
Better	4 (3.7%)	2 (6.3%)	

^+^ Chi-square test.

## Data Availability

The analyzed datasets of this study are available from the corresponding author on reasonable request.

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
