# Peer review of "Open Reduction in Subtrochanteric Femur Fractures Is Not Accompanied by a Higher Rate of Complications"

_medicina, 2021, doi:10.3390/medicina57070659_

Round 1

Reviewer 1 Report

A Seven day result seems not to be adaquat for this question. You Talk about the problem non Union, so a 6 month Observation time is to be requiered. Even a lot of serious complication will Not occur in the First week.

Author Response

Thank you very much for this helpful advice. You are right that a seven-day observation period is not the right period for the question of union of fractures. Union vs. non-union was not the primary outcome parameter of this study. We are sorry for this misunderstanding. We examined whether open-reduction of subtrochanteric fractures leads to more complications during inpatient treatment during inpatient treatment (median 17 days) and during the follow-up period of 120 days. Our study can therefore not make any statement about a possible union/non-union. We have therefore removed the term "union" from the conclusion.

The authors thank reviewers for their helpful comments and thoughts. We are extremely thankful for the suggestions and queries and hope the additional information leads to greater clarity and understanding of the presented investigation.

Reviewer 2 Report

Dear author, 

Thank you for the opportunity to revies this article. 

We understand that this manuscript assesses whether OR of subtrochanteric fractures in a certain age category is accompanied by a higher rate of complications than closed reduction.

"Introduction" defines the pathology and it's management problems among elder patients, and also the favoured surgical techniques for this type of fracture.

"Materials and methods" and "Results" are elaborate, and statistical analysis seems thorough.

In "Discussion" you should describe on what criteria you chose the adequate technique for each patient, whether it comprised open or closed reduction. We recommend CT scan with 3D reconstruction, eventually completed with 3D printing of the fracture site in order to optimise preop planning, as described in this reference: The use of 3D printing in improving patient-doctor relationship and malpractice prevention published in NEJM, DOI 10.4323/rjlm.2017.279

The field of interest pertaining a certain sensitive age group may bring value, especially for future literature reviews.

English needs improvement.

This article needs minor adjustments.

Author Response

Thank you very much for your helpful advices. Unfortunately, we are not able to reproduce which criteria lead the surgent to perform an open-reduction, since this is a registry study. We added this to the limitation section.

Furthermore, we added a statement regarding the possibility of a CT-scan with 3D reconstruction to our manuscript. (l.140-142). The possibility of making a 3D printed model does not seem to be possible in most clinics especially in the emergency department. 3D printing therefore seems to be currently rather reserved for individual complex cases in planned orthopedic surgery. It may, of course, as you write, lead to improved physician-patient bonding.

The manuscript was proofread again by an English native speaker.

The authors thank reviewers for their helpful comments and thoughts. We are extremely thankful for the suggestions and queries and hope the additional information leads to greater clarity and understanding of the presented investigation.

Round 2

Reviewer 2 Report

Dear authors,

We are delighted that you chose to mention 3D reconstruction in your manuscript, but it is necessary for you to cite the refference about CT scan followed by 3D reconstruction, a procedure that may become gold standard in the management of future related pathologies.

Author Response

Dear Reviewer,

thank you very much for your helpfull advice. As recommended we have added the citation.